# The Influence of Sugar Beet Cultivation Technologies on the Intensity and Species Biodiversity of Weeds

Barbora Kotlánová [1], Pavel Hledík [2], Stanislav Hudec [2], Petra Martínez Barroso [3], Magdalena Daria Vaverková [3,4], Martin Jiroušek [1] and Jan Winkler [1,*]

[1] Department of Plant Biology, Faculty of AgriSciences, Mendel University in Brno, Zemědělská 1, 613 00 Brno, Czech Republic; xkotlano@mendelu.cz (B.K.); jirosekk@mendelu.cz (M.J.)

[2] Crop Research Institute, Drnovská 507/73, 161 06 Prague, Czech Republic; hledik.pavel@vurv.cz (P.H.); cropscience@vurv.cz (S.H.)

[3] Department of Applied and Landscape Ecology, Faculty of AgriSciences, Mendel University in Brno, Zemědělská 1, 613 00 Brno, Czech Republic; xmarti15@mendelu.cz (P.M.B.); magdalena.vaverkova@mendelu.cz or magdalena_vaverkova@sggw.edu.pl (M.D.V.);

[4] Institute of Civil Engineering, Warsaw University of Life Sciences—SGGW, Nowoursynowska 159, 02 776 Warsaw, Poland

* Correspondence: jan.winkler@mendelu.cz

**Abstract:** Sugar beet production is highly affected by weeds. The structure of crop rotation, the use of intercrops and different tillage techniques bring several benefits to sugar beet cultivation and create different living conditions for weeds. The response of weed communities in sugar beet stands has not been studied. The experimental plot is in the cadastral area of Ivanovice na Hané (Czech Republic). During an eight-year monitoring period (2013–2020), 46 weed species were identified. The dominant species was *Chenopodium album*. There were also summer and winter weeds. A more varied crop rotation increased the intensity of weed infestation, with winter weeds being the most common. On the contrary, a higher proportion of cereals in the crop structure favors the presence of summer weeds. The tillage technology and the inclusion of catch crops did not significantly affect the intensity of weed infestation in sugar beet stands or the spectrum of weed species. Current cropping technologies have driven the evolution of weeds. Due to their short life cycles and relatively simple genomes, weeds can respond very quickly to technological measures and, thus, change their harmfulness.

**Keywords:** weed infestation; integrated weed control; cultivation technology; sugar beet; catch crop

## 1. Introduction

Sugar beet (*Beta vulgaris Altissima Group*) is an extremely sensitive crop to weed competition, especially in the early stages of growth. Effective weed control is a key factor for successful sugar beet production [1,2]. Weeds are the major biotic cause of yield loss in all field crops [3]. Herbicides are the basic tool for controlling weeds in sugar beet stands [4], but their use poses several risks. One of these risks is their toxicity. The most commonly used herbicides are not directly toxic to soil edaphon compared to some fungicides and insecticides [5–7], but the activity of edaphon is indirectly affected by the action of herbicides—when weed infestation density is abruptly reduced, food becomes limited, and shelter is lost for a number of organisms [8]. Because of the consequences of pesticide use, it is one of the most controversial aspects of modern agriculture at the individual, population and ecosystem levels [9,10]. The ecological effects of pesticides can last up to several years after their application [11], which is why other weed control methods are also used. Tillage is another tool used to control weeds. When growing sugar beets, it is possible to use conservation tillage [12]. Conservation tillage includes no-tillage (NT) and minimum tillage (MT). These practices have a beneficial effect on reducing the rate of soil organic carbon (SOC) mineralization, soil water retention [13–15], reducing water

runoff and erosion [16], and stabilizing nutrient uptake [2]. Soil edaphon activity benefits from conservation tillage owing to the retention of food on the soil surface and the reduction of physical soil disturbance [17]. Despite these benefits, many farmers worldwide have not yet adopted conservation agriculture practices [18,19]. A commonly cited concern is the difficulty of weed control—no-tillage technology provides a limited direct weed control capability, especially where herbicide-resistant weed populations predominate [20,21].

Ploughing, as part of traditional tillage, plays a key role in creating a favorable soil environment and promoting crop growth and has a beneficial effect on soil moisture and temperature. Ploughing crop residues into the soil increases the organic matter content, which affects the quality of future yields [22]. The yield and quality of sugar beet are influenced by many environmental and agronomic factors [23]; therefore, new procedures for their cultivation, which would preserve the optimal state of the soil environment, are being sought [16]. Traditional tillage is replaced by undermining, loosening, and mulching [24]. According to studies carried out in the United States, strip tillage technologies are used for sugar beets, and yields are similar to those produced by traditional cultivation [25,26]. According to Górský et al. [27], the selected tillage technology markedly affects the yield of sugar beet tubers, whereas the sugar content is more influenced by the variety. However, both yield and sugar content were the most affected by the year. Arvidsson et al. [28] point out that soil compaction has a significant effect on sugar beet yield; therefore, it is essential to care for the good physical condition of the soil. Soil compaction is enhanced by the number of crossings in the agricultural machinery. If the crossings of the machinery are limited, sugar beet yield can increase by up to 10% [29]. Soil degradation caused by intensive cultivation affects the sustainability of agroecosystems. Therefore, much attention is being paid to integrated soil health care strategies that use agronomic practices such as crop rotation, catch crops and conservation tillage [30–33].

Crop rotation is considered an environmentally friendly approach that diversifies farming systems, reduces the intensity of pressure on the agricultural ecosystem, and prevents the action of specific harmful organisms, including weeds. Previous research has shown that crop rotation in more biodiverse agricultural systems provides substantial environmental benefits [34] and can effectively reduce weeds. This results in decreased yield loss owing to weeds. Some crops suppress weed growth better than others. Several studies have shown that crop rotation systems and the selection of suitable crops can effectively regulate weed intensity [35,36]. Crop rotation alleviates negative crop stress, ensuring a higher yield than monoculture cropping [37].

Other important measures affecting agrosystems are catch crops, which have the potential to ensure sustainable agriculture, support production diversification, reduce pest and pathogen pressure [38–40], improve soil health, optimize resource use [41,42] and increase biodiversity in fields [43]. Multispecies mixtures of catch crops maximize land use and increase biodiversity [44]. The most common use of catch crop biomass is green fertilization, which increases the soil organic matter content. If leguminous crops are among the catch crops, green fertilization enables nitrogen fixation [45]. Most studies analyzing the effects of growing crop systems on weed infestation are related to cereals, including maize [46–52]. Scientific attention has also been paid to evaluating the possibility of using reduced tillage for legumes, including soy [53–56], or oil crops such as rapeseed [57]. The focus on other types of crops is omitted.

The different structures of crop rotation, use of catch crops, and different tillage technologies provide several benefits in the field of sugar beet cultivation. Agricultural practices in the cultivation of sugar beets affect the living conditions of weed populations. The reaction of weed populations in sugar beet stands are unknown. According to our hypothesis, different tillage practices, a higher proportion of cereals in crop rotation and the inclusion of catch crops increases the biodiversity of field weeds. The sub-goals leading to the confirmation of our hypothesis are (i) to determine the effect of the share of cereals in the sowing procedure on the intensity and species composition of weeds, (ii) to determine the effect of the inclusion of a catch crop and different tillage systems on the intensity and

species composition of weeds and (iii) to analyze the species composition of weeds growing with sugar beets. Describing the relationships between sugar beet cultivation technologies and weed populations will enable the optimization of the potential of individual cultivation interventions for weed control.

## 2. Materials and Methods

### 2.1. Characteristics of the Experimental Plot

The experimental plots are located within the premises of the field trial station in the cadastral territory of the municipality of Ivanovice na Hané (GPS 49.3123383 N, 17.0948131 E), in the eastern Czech Republic. The field trial station was established in 1989 by the Plant Production Research Institute in Prague Ruzyně. The terrain is mostly flat to slightly sloped. The average altitude is approximately 230 m above sea level. The long-term average annual precipitation is 564 mm, and the long-term average temperature is 8.6 °C. Long-term rainfall and temperature data for each month are shown in Table 1.

**Table 1.** Long-term averages of temperature and average monthly precipitation for each month (1961 to 1990).

| Month | I. | II. | III. | IV. | V. | VI. | VII. | VIII. | IX. | X. | XI. | XII. |
|---|---|---|---|---|---|---|---|---|---|---|---|---|
| Precipitation (mm) | 25 | 27 | 33 | 44 | 63 | 79 | 74 | 69 | 37 | 39 | 44 | 30 |
| Temperatures (°C) | −2.6 | −1.2 | 3.9 | 9.0 | 14.6 | 16.6 | 18.7 | 18.3 | 14.4 | 9.3 | 2.8 | −0.3 |

The experimental land was formed by a loamy chernozem soil type and was heavily leached. The depth of the topsoil was 0.4 m, the pH was 7.1, and the humus content was 4.39%. The available nutrient contents were P 177 mg/kg, K 573 mg/kg, Mg 215 mg/kg and Ca 3988 mg/kg.

### 2.2. Description of the Field Trial

Sugar beets were grown within a crop rotation system that had 3 variants, each with different proportions of cereals. The first sowing procedure (I.) had a 33.3% share of cereals. The crop rotation had the following structure: alfalfa (first and second productive years), winter wheat, corn for silage, sugar beet and spring barley. The second sowing procedure (II.) had a 50.0% share of cereals. Crops were grown in the following order: peas, silage corn, winter wheat, winter wheat, sugar beet and spring barley. The third crop procedure (III.) had a 66.6% share of cereals and the crops were grown in the following order: winter wheat, peas, winter wheat, spring barley, sugar beets and spring barley.

Tillage systems were another factor in the field trial. The tillage technology had 4 variants: (i) traditional tillage (NoIC2T), which consisted of a first ploughing to a depth of 0.22 m and a second ploughing to a depth of 0.28 m, (ii) ploughing and loosening (NoIC1T), consisting of a ploughing depth of 0.22 m and subsequent loosening, (iii) traditional tillage with intercropping (IC2T), which consisted of two plowings to depths of 0.22 and 0.28 m, subsequent leveling of the soil and sowing of a freezing catch crop, (iv) ploughing, loosening with catch crops (IC1T), which was formed by ploughing to a depth of 0.22 m with subsequent loosening and sowing of the freezing catch crop. As part of the field trial, *Phacelia tanacetifolia* was used as the catch crop.

Sugar beet cultivation technology was the same for all variants. Spring tillage before sowing was performed using a combine harvester and sowing with a precision seed drill. Sugar beet was sown at an inter-row distance of 0.45 m with a plant spacing of 0.18 m. Sugar beet was fertilized with 40 t/ha of manure, 50 kg/ha of P and 110 kg/ha of K before ploughing in the spring during pre-sowing. A soil preparation with 60 kg/ha of N and 60 kg/ha N was used for fertilization during vegetation.

Between 2013 and 2018, the variety Marenka was planted in the trial. Between 2019 and 2020, the variety KIPUNJI SMART was planted in the trial. The pesticides used were

Sumi Alpha insecticide (esfenvalerate) at 0.1 L/ha and Tern fungicide (fenpropidin) at 0.5 L/ha. The herbicides used are listed in Table 2.

**Table 2.** Specific dates for the evaluation of weeds in sugar beet stands.

| Year | Date of Application | Herbicide (Active Ingredient; Dose) |
|---|---|---|
| 2013–2016 | At the stage of two real sugar beet leaves | Mix Double EC (desmedifam, fenmedifam; 1.25 L/ha) |
| | At the stage of four real sugar beet leaves | Mix Double EC (desmedifam, fenmedifam; 1.5 L/ha) |
| 2017–2018 | At the stage of two real sugar beet leaves | Pyramin Turbo (chloridazon; 1.6 L/ha) |
| | At the stage of four real sugar beet leaves | Pyramin Turbo (chloridazon; 1.6 L/ha) |
| 2019–2020 | At the stage of two real sugar beet leaves | Conviso ONE (foramsulfuron, thiencarbazone-methyl; 0.5 L/ha) |
| | At 10–15 days after the first application | Conviso ONE (foramsulfuron, thiencarbazone-methyl; 0.5 L/ha) |

*2.3. Evaluation of Weed Infestation*

Weed infestation of sugar beet stands was evaluated between 2013 and 2020 using a numerical method. The number of weed individuals was determined over an area of 1 m$^2$ in 12 repetitions for each variant of crop rotation, tillage, and year. The assessment was performed each spring before the application of herbicides. The specific date of the evaluation is given in Table 3, and the evaluation was always performed at the same growth stage, when the sugar beet plants had 2–4 true leaves. The taxonomic nomenclature of the plants was determined according to the method described by Kaplan et al. [58].

**Table 3.** Specific dates for evaluating weeds in sugar beet stands.

| Year | Date of Evaluation |
|---|---|
| 2013 | 30 April–1 May |
| 2014 | 2 May–4 May |
| 2015 | 3 May–5 May |
| 2016 | 7 May–9 May |
| 2017 | 13 May–15 May |
| 2018 | 12 May–13 May |
| 2019 | 27 April–30 April |
| 2020 | 18 May–20 May |

The weed species found were divided into biological groups according to Jursik et al. [59]. The first group, called spring weeds, germinates mainly in early to late spring. The germinating weeds tolerate temperatures below 0 °C only for a short time and only exceptionally survive the winter period. The second group is known as summer weeds, which germinate mainly from spring to summer when there is sufficient moisture, as they do not survive temperatures below 0 °C. The third group is called winter weeds; they germinate in autumn and also in spring until summer if there is sufficient moisture, and they survive temperatures below 0 °C and are able to overwinter. The fourth group is called perennial and other weeds. The perennial species have more growing seasons and undesirable crops have been included in the other weeds.

The differences in the number of weed individuals between the crop rotation variants, tillage treatments, and years were statistically evaluated using the Kruskal–Wallis non-parametric test with the Nemenyi post hoc statistical test. Both tests were conducted and boxplots were generated using the R programming language [60], and the PMCMRplus package [61] was used for Nemenyi test.

The representation of individual weed species across different crop rotation variants, tillage methods and monitoring years was further processed using multivariate analysis of the ecological data. The selection of the optimal analysis depended on the length of the gradient, as determined by segmental detrended correspondence analysis (DCA). Subsequently, canonical correspondence analysis (CCA) was performed. Statistical significance was determined using the Monte Carlo test, in which 999 permutations were calculated. The data were processed using the computer program Canoco 5 [62].

### 3. Results

During the eight-year long monitoring, 46 species of weeds were found. The average number of weed individuals and the representation of the biological groups of weeds are shown in Figures 1–4. The results showed that crop rotation, use of catch crops, and tillage had a significant effect on the intensity of weed infestation (Figure 5). Variant IC1T creates conditions for statistically proven higher weed infestation. The differences between the other variants were not statistically significant. The intensity of weed infestation was also significantly influenced by the year. As shown in Figure 6, the level of weed infestation intensity was more balanced in crop rotation according to sowing method I. By contrast, more extremes in weed infestation intensity were found in crop rotation according to variant III.

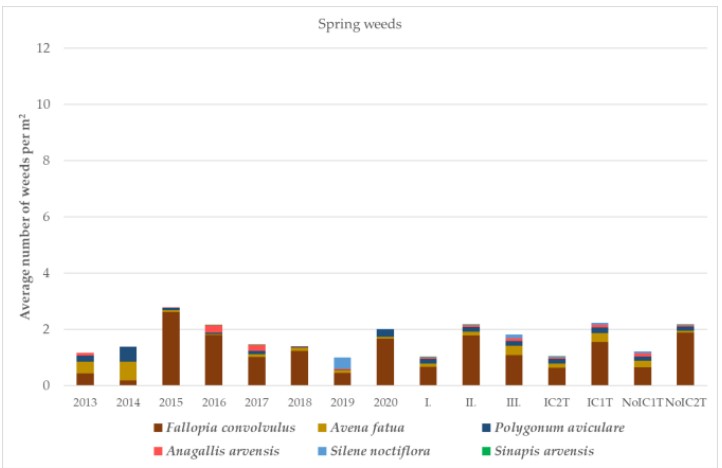

**Figure 1.** Representation of weed species (spring weeds) individual variants of the field trial in the monitored years.

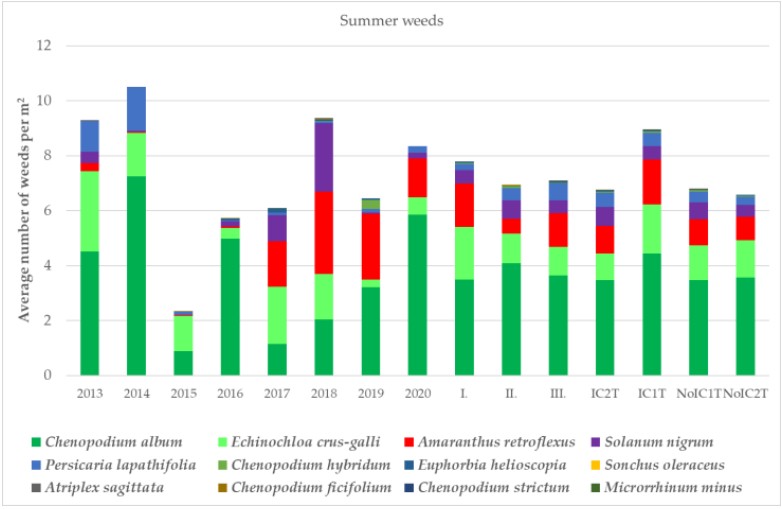

**Figure 2.** Representation of weed species (summer weeds) individual variants of the field trial in the monitored years.

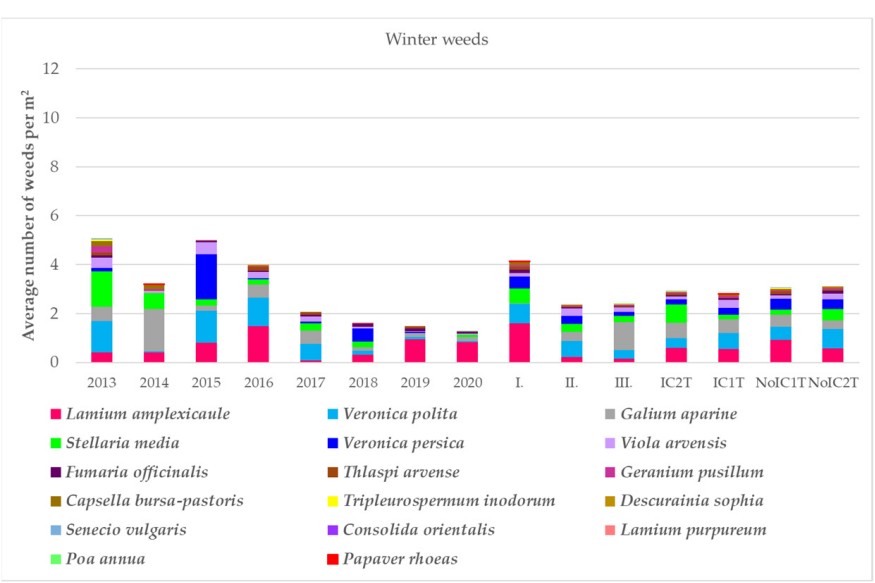

**Figure 3.** Representation of weed species (winter weeds) individual variants of the field trial in the monitored years.

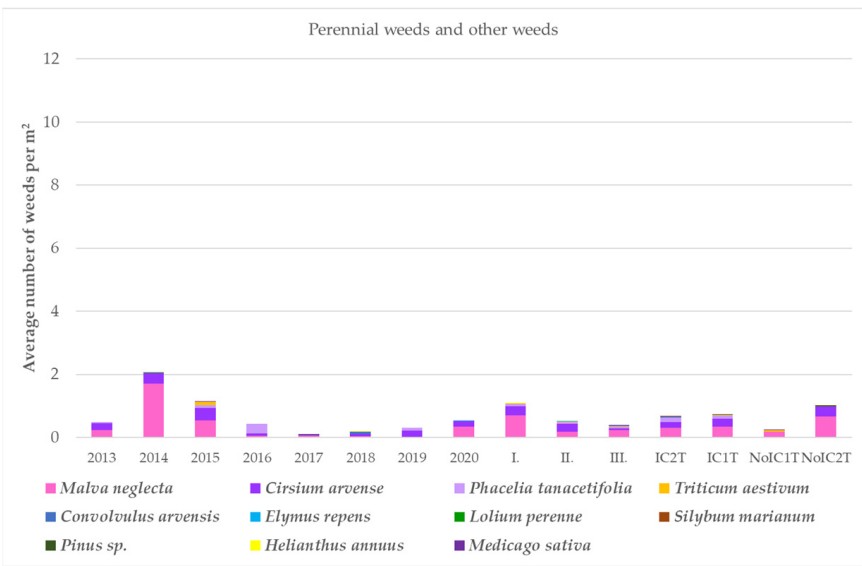

**Figure 4.** Representation of weed species (perennial weeds and other weeds) individual variants of the field trial in the monitored years.

Crop rotation, the use of catch crops and tillage also have a significant effect on the weed species structure in sugar beets. The results of the CCA, which evaluated the representation of individual weed species in different crop rotation variants, were significant at the $\alpha = 0.001$ significance level for all canonical axes. A graphical representation of the CCA analysis results is displayed in Figure 7. Based on CCA analysis, the weed species found could be divided into four groups. The first group occurred mainly in variant I. The group constitutes of following species *Amaranthus retroflexus* (*AmaRetr*), *Capsella bursa-pastoris* (*CapBurs*), *Cirsium arvense* (*CirArve*), *Descurainia sophia* (*DesSoph*), *Echinochloa crus-galli* (*EchCrus*), *Fumaria officinalis* (*FumOffi*), *Geranium pusillum* (*GerPusil*), *Helianthus annuus* (*HelAnnu*), *Chenopodium ficifolium* (*CheFici*), *Chenopodium strictum* (*CheStri*), *Lamium amplexicaule* (*LamAmpl*), *Lolium perenne* (*LolPere*), *Malva neglecta* (*MalNegl*), *Papaver rhoeas* (*PapRhoe*), *Phacelia tanacetifolia* (*PhaTana*), *Sinapis arvensis* (*SinArve*), *Stellaria media* (*SteMedi*), *Thlaspi arvense* (*ThlArve*), *Veronica persica* (*VerPers*) and *Veronica polita* (*VerPoli*).

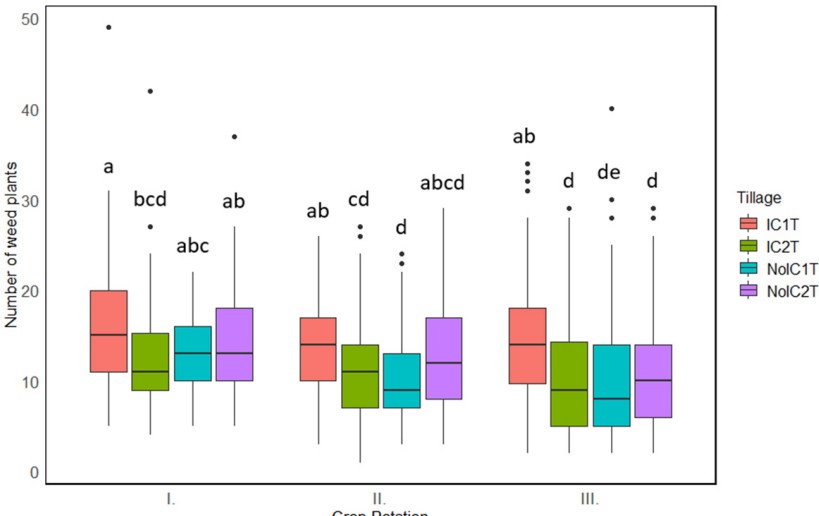

**Figure 5.** Intensity of weed infestation in conditions of different crop rotation variants (I. = sugar beet after corn, II. = sugar beet after winter wheat, III. = sugar beet after spring barley) and for individual tillage technologies (IC = intercrop, NoIC = without intercrop, 1T = single ploughing with subsequent loosening, 2T = double ploughing). Boxplots present median and inter-quartile range (IQR; 25–75% range of data), whiskers show 1.5 × IQR and finally black spots are outliers. Different letters (a–e, with decreasing median) above boxplots indicate significant difference calculated by the Nemenyi post hoc test ($p < 0.05$).

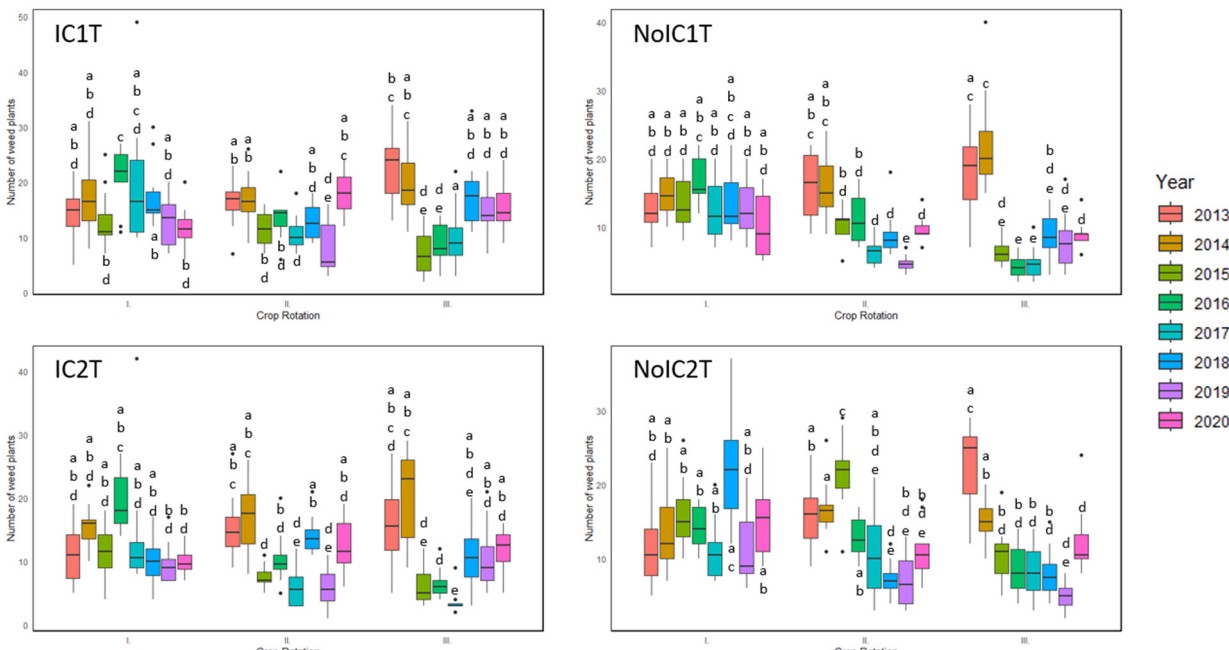

**Figure 6.** Intensity of weed infestation in conditions of different crop rotations and for individual tillage technologies in the monitored years. Boxplots present median and inter-quartile range (IQR; 25–75% range of data), whiskers show 1.5 × IQR and finally black spots are outliers. Different letters (a–e, with decreasing median) above boxplots indicate significant difference calculated by the Nemenyi post hoc test ($p < 0.05$).

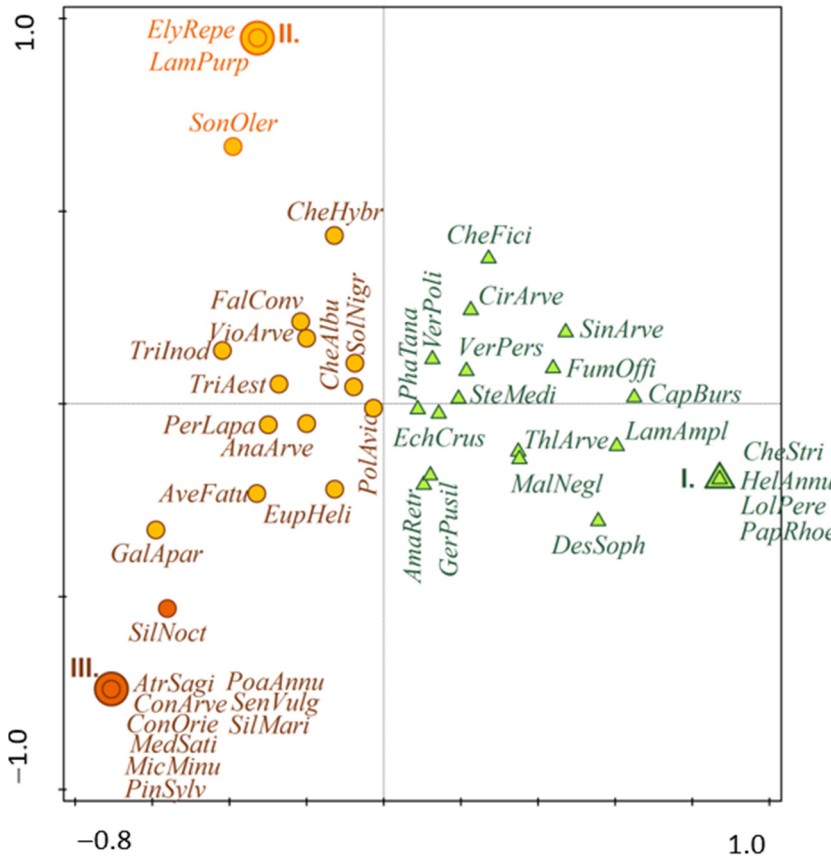

**Figure 7.** Relationship between weed species and crop rotation—results of CCA analysis (total explained variability = 11.8%; F ratio = 15.9; *p*-value = 0.001).

The second group of species occurred mainly in variant II. The species are *Elymus repens* (*ElyRepe*), *Lamium purpureum* (*LamPurp*) and *Sonchus oleraceus* (*SonOler*).

The third group of species occurred mainly in variants II. and III. The species are the following: *Anagallis arvensis* (*AnaArve*), *Avena fatua* (*AveFatu*), *Euphorbia helioscopia* (*EupHeli*), *Fallopia convolvulus* (*FalConv*), *Galium aparine* (*GalApar*), *Chenopodium album* (*CheAlbu*), *Chenopodium hybridum* (*CheHybr*), *Persicaria lapathifolia* (*PerLapa*), *Polygonum aviculare* (*PolAvic*), *Solanum nigrum* (*SolNigr*), *Triticum aestivum* (*TriAest*), *Tripleurospermum inodorum* (*TriInod*) and *Viola arvensis* (*VioArve*).

The fourth group of species occurred mainly in variant III. The species are the following: *Atriplex sagittata* (*AtrSagi*), *Convolvulus arvensis* (*ConArve*), *Consolida orientalis* (*ConOrie*), *Medicago sativa* (*MedSati*), *Microrrhinum minus* (*MicMinu*), *Pinus sylvestris* (*PinSylv*), *Poa annua* (*PoaAnnu*), *Senecio vulgaris* (*SenVulg*), *Silybum marianum* (*SilMari*) and *Silene noctiflora* (*SilNoct*).

The results of the CCA, which assessed the representation of individual weed species in individual variants of the use of catch crops and tillage, were significant at the $\alpha = 0.001$ significance level for all canonical axes. A graphical representation of the CCA results is shown in Figure 8. Based on CCA analysis, the weed species found could be divided into five groups.

The first group of species occurred mainly in the IC2T variant. This group consists of species *Convolvulus arvensis* (*ConArve*), *Chenopodium ficifolium* (*CheFici*), *Phacelia tanacetifolia* (*PhaTana*) and *Sonchus oleraceus* (*SonOler*).

The second group of species occurred mainly in the IC1T variant. This group consists of species *Atriplex sagittata* (*AtrSagi*), *Consolida orientalis* (*ConOrie*), *Microrrhinum minus* (*MicMinu*), *Papaver rhoeas* (*PapRhoe*), *Pinus sylvestris* (*PinSylv*) and *Poa annua* (*PoaAnnu*).

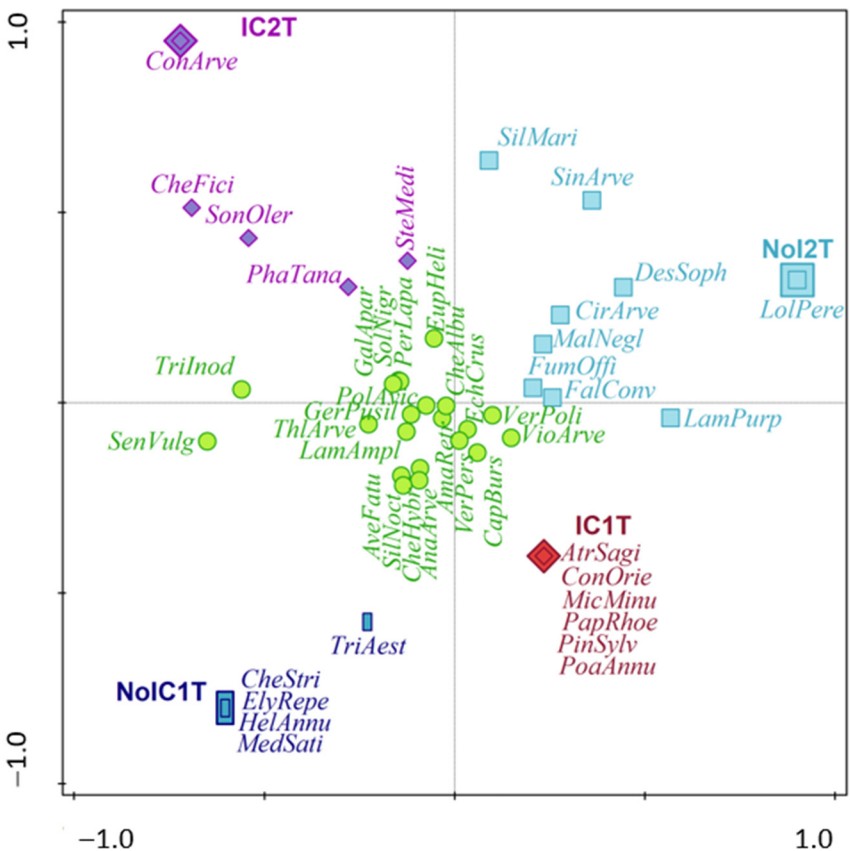

**Figure 8.** Relationship between identified weed species and tillage technologies—results of CCA analysis (total explained variability = 11.8%; F ratio = 15.9; *p*-value = 0.001).

The third group of species occurred mainly in the NoI2T variant group. This group consists of species *Cirsium arvense* (*CirArve*), *Descurainia Sophia* (*DesSoph*), *Fallopia convolvulus* (*FalConv*), *Fumaria officinalis* (*FumOffi*), *Lamium purpureum* (*LamPurp*), *Lolium perenne* (*LolPere*), *Malva neglecta* (*MalNegl*), *Silybum marianum* (*SilMari*) and *Sinapis arvensis* (*SinArve*).

The fourth group of species occurred mainly in the NoIC1T variant. This group consists of species *Elymus repens* (*ElyRepe*), *Helianthus annuus* (*HelAnnu*), *Chenopodium strictum* (*CheStri*), *Medicago sativa* (*MedSati*), *Stellaria media* (*SteMedi*) and *Triticum aestivum* (*TriAest*).

The fifth group of species was made up of species that did not have any preference for any of the variants and whose occurrence was influenced more by other factors. Those were following species: *Amaranthus retroflexus* (*AmaRetr*), *Anagallis arvensis* (*AnaArve*), *Avena fatua* (*AveFatu*), *Capsella bursa-pastoris* (*CapBurs*), *Echinochloa crus-galli* (*EchCrus*), *Euphorbia helioscopia* (*EupHeli*), *Galium aparine* (*GalApar*), *Geranium pusillum* (*GerPusil*), *Chenopodium album* (*CheAlbu*), *Chenopodium hybridum* (*CheHybr*), *Lamium amplexicaule* (*LamAmpl*), *Persicaria lapathifolia* (*PerLapa*), *Polygonum aviculare* (*PolAvic*), *Senecio vulgaris* (*SenVulg*), *Silene noctiflora* (*SilNoct*), *Solanum nigrum* (*SolNigr*), *Thlaspi arvense* (*ThlArve*), *Tripleurospermum inodorum* (*TriInod*), *Veronica persica* (*VerPers*), *Veronica polita* (*VerPoli*) and *Viola arvensis* (*VioArve*).

The results of the CCA, which evaluated the representation of individual weed species in the monitored years, were significant at the $\alpha$ = 0.001 significance level for all canonical axes. A graphical representation of the CCA results is shown in Figure 9. Based on CCA analysis, the weed species found could be divided into three groups.

The first group of species occurred mainly in 2013, 2014, 2016 and 2020. This group consists of species *Anagallis arvensis* (*AnaArve*), *Atriplex sagittata* (*AtrSagi*), *Avena fatua* (*AveFatu*), *Capsella bursa-pastoris* (*CapBurs*), *Cirsium arvense* (*CirArve*), *Consolida orientalis* (*ConOrie*), *Echinochloa crus-galli* (*EchCrus*), *Elymus repens* (*ElyRepe*), *Fallopia convolvulus* (*FalConv*), *Galium aparine* (*GalApar*), *Geranium pusillum* (*GerPusil*), *Chenopodium album* (*CheAlbu*), *Lamium amplexicaule* (*LamAmpl*), *Lolium perenne* (*LolPere*), *Malva neglecta* (*MalNegl*), *Papaver rhoeas* (*Pa-*

pRhoe), *Persicaria lapathifolia* (*PerLapa*), *Phacelia tanacetifolia* (*PhaTana*), *Poa annua* (*PoaAnnu*), *Polygonum aviculare* (*PolAvic*), *Senecio vulgaris* (*SenVulg*), *Sonchus oleraceus* (*SonOler*), *Stellaria media* (*SteMedi*), *Thlaspi arvense* (*ThlArve*), *Tripleurospermum inodorum* (*TriInod*), *Veronica polita* (*VerPoli*) and *Viola arvensis* (*VioArve*).

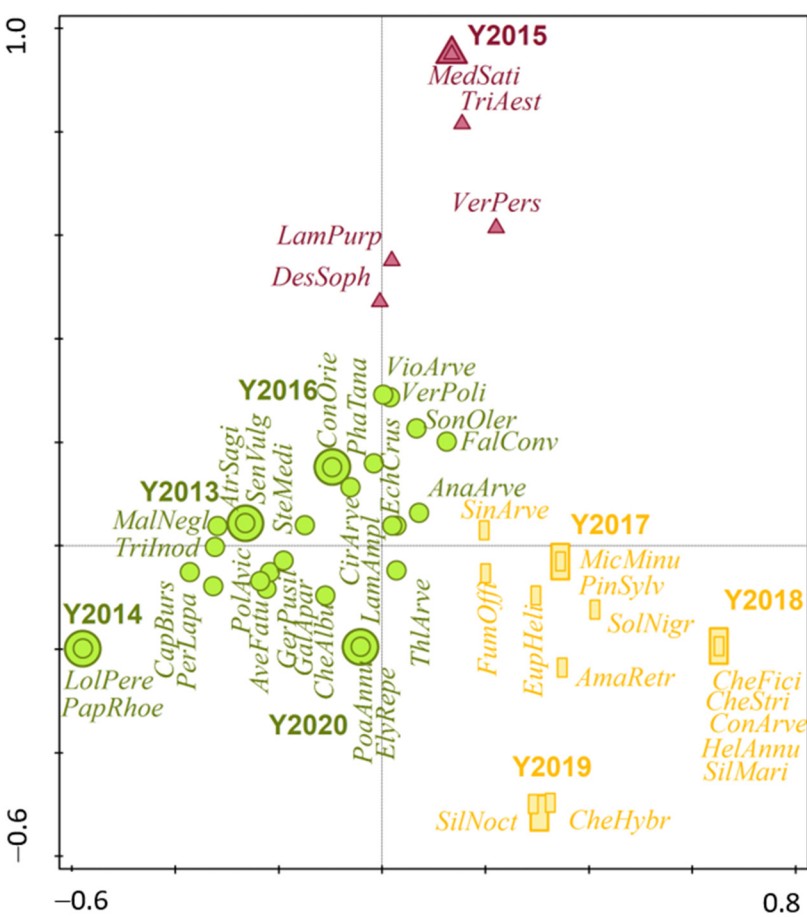

**Figure 9.** Relationship between identified weed species and evaluated years—results of CCA analysis (total explained variability = 11.8%; F ratio = 15.9; *p*-value = 0.001).

The second group of species occurred mainly in 2015. This group consists of species *Descurainia sophia* (*DesSoph*), *Lamium purpureum* (*LamPurp*), *Medicago sativa* (*MedSati*), *Triticum aestivum* (*TriAest*) and *Veronica persica* (*VerPers*).

The third group of species occurred in 2017, 2018 and 2019. This group consists of *Amaranthus retroflexus* (*AmaRetr*), *Convolvulus arvensis* (*ConArve*), *Euphorbia helioscopia* (*EupHeli*), *Fumaria officinalis* (*FumOffi*), *Helianthus annuus* (*HelAnnu*), *Chenopodium ficifolium* (*CheFici*), *Chenopodium hybridum* (*CheHybr*), *Chenopodium strictum* (*CheStri*), *Microrrhinum minus* (*MicMinu*), *Pinus sylvestris* (*PinSylv*), *Silybum marianum* (*SilMari*), *Silene noctiflora* (*SilNoct*), *Sinapis arvensis* (*SinArve*) and *Solanum nigrum* (*SolNigr*).

## 4. Discussion

The most common tool for weed control is the widespread application of herbicides, which raises concerns regarding their impact on the environment. To reduce the amount of herbicides applied, principles of integrated plant protection are being developed to employ other cultivation methods (crop rotation, intercrop rotation, and tillage) to reduce weed infestation. The goal of integrated plant protection is to reduce the weed population to an acceptable level and limit the impact of herbicides on soil quality, water, and other non-target organisms [63,64].

Despite the use of all control practices, it is surprising that weed frequency increased on average. For example, in Denmark, there was a 25% increase in weeds in intensively cultivated sugar beet stands when comparing two periods, 1987 to 1989 and 2001 to 2004. The main reason for this increase was most likely a change in the spectrum of herbicides used. Although the average frequency of weeds has increased, substantial changes have occurred mainly in certain dominant weed species [65]. Thus, weeds remain an important factor limiting yield and complicating sugar beet cultivation.

In the monitored sugar beet stands, weeds from the group of summer weeds, especially the species *Chenopodium album*, were mainly present. They accounted for 29.7% of all weed individuals. There were also the species *Echinochloa crus-galli* (10.7%) and *Amaranthus retroflexus* (8.8%). Regarding the group of spring weeds, the species *Fallopia convolvulus* had the highest share (9.4%), and regarding the group of winter weeds, the species *Lamium amplexicaule* (5.3%) and *Veronica polita* (4.7%) were the most abundant. Sugar beet stands create conditions that enable the establishment of weed species with similar requirements (*Chenopodium spp.*). These species occupy a dominant position and represent the main competition for sugar beet. Bojarszczuk et al. [66] report that *Chenopodium album* is among the most abundant weeds in all tillage systems. The trend of the appearance of the dominant type of weed is manifested in the weed infestation of sugar beet, i.e., species with marked representation compared to other species. In the Czech Republic, this species is also *Chenopodium album*, which infests stands treated with herbicides [67].

According to our results, crop rotation is the cultivation intervention with the highest impact. The importance of crop rotation in modern agriculture lies in the harmonization of relationships between human activities and the ecosystems that provide for us [68]. Ramsdale et al. [69] report that according to their results, it is possible to reduce weed density and limit herbicide application by using an appropriate combination of crop rotation and tillage. However, it is not possible to exclude the use of herbicides for dominant weed control if heavy weeding causing high yield losses can be prevented.

A varied seeding procedure (I.) creates more favorable conditions for the establishment of weeds with different biological requirements than the ones of sugar beet. These are mainly species from the group of winter weeds. The lower frequency of cereals (33.3%) led to a more balanced intensity of sugar beet weed infestation in the monitored period. According to Saulic et al. [70] crop rotations reduce weed seed stocks; however, it is a prerequisite to prevent the infestation of weed seeds originating from manure application, which is a source of weed seeds in any crop rotation. Manure, as a source of weeding, was also pointed out by Winkler et al. [71]. The sugar beet stands evaluated were fertilized with manure, in which the presence of weed seeds could be assumed. The weed seeds in the manure probably influenced the intensity and species composition of weed infestation of the evaluated sugar beet stands.

A higher proportion of cereals in the cultivation procedures (50.0% and 66.6%) supports the occurrence of species from the group of summer weeds and a higher representation of the dominant species *Chenopodium album*. Similar timing of herbicide application, sowing and harvesting are favorable for the occurrence of dominant weed species. Crop rotations with similar growing seasons maintained the dominance of certain weed species. When crop rotation was limited, a more extreme weeding intensity was recorded. A higher proportion of cereals probably increased the heterogeneity of weed infestation of the subsequently grown sugar beet. This higher heterogeneity can be reflected in the extreme differences in the weed infestation of sugar beet. Years when sugar beet stands were subjected to an extremely high intensity of weed infestation had a negative impact on subsequent weed control, which can result in an increased application of herbicides. Herbicides remain an essential tool in weed control and are applied according to strategies that reflect the composition of the local weed population [8]. The effectiveness of herbicide strategies is of fundamental importance for the weed control of sugar beets [72].

The intensity of weed infestation and the species spectrum of weeds were not significantly influenced by tillage technology or the inclusion of catch crops. Only the variant

with ploughing and loosening with the inclusion of catch crop (IC1T) created conditions for higher weed infestation. The differences between the other variants were not statistically significant. These results are consistent with a number of studies reporting inconclusive effects of different tillage systems on weeds [73–75]. Nevertheless, many studies have focused on to the fact that tillage systems affect weed vegetation, fragmentation of rhizomes and vegetative reproductive organs, and distribution of weed seeds in the soil profile, but also affect the soil environment, such as soil aeration [76–79].

Soil processing and the inclusion of catch crops will probably be more important for sugar beet stands from the point of view of the influence on the physical, chemical and biological properties of the soil, which directly affect the growth and development of crops and their quantity and yield quality [48,80–82]. Correctly executed soil processing primarily prevents its compaction [83,84]. Sugar beet root growth is influenced by soil structure [16,85]. Higher soil compaction reduces the growth of fibrous roots [86], which limits emergence rate and reduces the leaf area of sugar beet. The consequences of the aforementioned are reduced yield and sugar content of tuber [87,88]. Tillage affects the amount of labor and energy costs and also the state of the soil environment [89], water erosion, temperature fluctuations and soil organic matter content [17]. These are decisive factors for the application of different tillage technologies under practical conditions.

The use of catch crops is an ecologically acceptable weed control practice because catch crops enhance the competitiveness of the main crops against weeds. They are used in organic and low-input agriculture [90,91]. Catch crops are able to suppress weeds, but the degree of regulation depends on the choice of catch crop species, their share in the mixture and the density of the catch crop stand [92]. Catch crop residues can directly regulate the germination and growth of weed species in the main crop [93]. However, based on our results, it is evident that the regulatory ability of catch crops is not manifested in the subsequently grown sugar beets.

Catch crops provide diverse ecosystem services that maintain and improve soil and water quality, reduce erosion, and increase soil organic matter content. Catch crops reduce nitrogen leaching and the cost of protection against other harmful agents [94–97]. In the long term, many ecosystem services provided by the employment of catch crops can improve the resilience of the main crop with positive feedback on yield stability and the reduction of external input requirements [98]. Based on the results of multivariate analyses, it can be stated that in certain years, the species composition of weeds was similar. The meteorological conditions in the monitored years had a statistically significant effect on the occurrence of weeds. The responses of the individual weed species to changes in meteorological conditions were different. Based on a literature survey [99,100], it is clear that moisture and low temperatures, accompanied by the stratification process, disrupt or strengthen the dormant state of seeds and consequently affect the germination and emergence of weeds. The long-term collection of detailed meteorological data and monitoring of weed infestation in a given location could be used to forecast weed infestation in the following growing season.

Current crop production technologies interact with weeds, which are fundamental drivers of weed evolution. The evolutionary history and ecological dynamics of weeds are intrinsically linked to human activity (agriculture) [101]. Because of their short life cycles and relatively simple genomes, weed species can very quickly change their responses to technological measures, and thus, their harmfulness.

The function of weed vegetation in agroecosystems can be considered from different perspectives. The harmfulness of weeds is the dominant perceived quality because weeds compete with cultivated crops. The damage caused by weeds affects the economics of agricultural production and yield. The current globalization of the economy, climate change and human population growth will increase the pressure on weed control. The possibility of predicting weed infestation based on the course of the weather is completely omitted in the current weed control strategies. Studying the effect of weather on weeds can lead to the formulation of specific links and relationships between the course of the weather

and the occurrence of weeds. This issue faces several scientific barriers, such as the lack of continuous monitoring of weeds' occurrence and the course of the weather during long-term experiments. Another major complication is the high variability of weed populations, which limits the statistical evaluation of populations. Creating a model for forecasting future weed infestation would enable farmers to make weed control more efficient, which would facilitate better and more timely mechanical weed control. This may result in a reduction in herbicide use. A new method of weed control must be implemented under conditions of sustainable agricultural intensification.

## 5. Conclusions

Sugar beet cultivation technologies affect the intensity and species composition of weed infestation. *Chenopodium album* was the dominant weed species in all the treatments. Summer and winter weed species were abundant among the weed groups. A more diverse crop rotation increases the intensity of weed infestation and makes the group of winter weeds more abundant. In contrast, a higher proportion of cereals in the crop structure favors the representation of summer weeds and the dominant species, *Chenopodium album*. The division of weed species into groups leads to some generalizations that may obscure some of the characteristics of the weeds. From this point of view, it is necessary to study the biology of weeds and their characteristics. Of particular importance is the germination temperature of weed seeds and also the ability to survive frost. Based on the new facts, the classification of weeds into modified biological groups could be adjusted.

The tillage technique and the inclusion of a catch crop did not significantly affect the intensity of weeding and the species spectrum of weeds in sugar beet stands. These results are in agreement with previous studies. Tillage and the effect of catch crops need to be evaluated primarily in terms of their impact on the soil environment and ecosystem functions. Understanding the relationship between sugar beet cropping technologies and weed populations will enable us to better exploit the potential of individual cropping practices for weed control.

**Author Contributions:** Conceptualization, J.W., B.K. and P.H.; methodology, J.W., B.K. and P.H.; validation, J.W. and B.K.; formal analysis, J.W., M.J. and B.K.; investigation, J.W., S.H. and B.K.; resources, J.W. and B.K.; data curation, J.W. and B.K.; writing—original draft preparation, J.W. and B.K.; writing—review and editing, J.W., P.M.B., M.D.V. and B.K.; visualization, J.W. and B.K.; supervision, J.W., M.D.V. and P.H.; project administration, J.W. and S.H.; funding acquisition, J.W. and B.K. All authors have read and agreed to the published version of the manuscript.

**Funding:** The Crop Research Institute, which focuses on long-term experiments, supports the work.

**Data Availability Statement:** Not applicable.

**Conflicts of Interest:** The authors declare no conflicts of interest.

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
