# Peer review of "The Influence of Sugar Beet Cultivation Technologies on the Intensity and Species Biodiversity of Weeds"

_agronomy, doi:10.3390/agronomy14020390_

Round 1

Reviewer 1 Report

Comments and Suggestions for Authors

Review of the manuscript entiled ” The influence of sugar beet cultivation technologies on the intensity and species biodiversity of weeds”

The reviewed manuscript concerns the impact of crop rotation, tillage technology and the inclusion of catch crops on the number and species composition of weeds infesting sugar beet cultivation. The reviewed article is a valuable addition to knowledge on this issue. As expected, the conclusions are not clear, which results from the number of factors influencing weed infestation and their mutual interaction. The research methodology does not raise any serious concerns, but it should be detailed in section “2.3. Evaluation of weed infestation”. Information "The weed assessment was performed each spring before the application of herbicides”(l. 146-147) should be more detailed and cover the month and decade of assessment. This is a certain time interval, because the date of beet sowing, and therefore also of weed control, probably differed between years. The mere statement that the presence or even dominance of summer weeds was detected during the spring assessment raises doubts. The advantage of the manuscript is the careful and accurate processing of the results using CCA. The most controversial issue is the division of weeds into four groups (spring, summer, winter, perennials and others) according to biology, with particular emphasis on the germination period. Assigning weeds to the groups mentioned is not always correct.

The group of spring weeds was most correctly determined, although Polygonum aviculare seedlings develop both in spring and summer. Among the weeds identified as summer, the seeds of Chenopodium hybridum germinated rather in spring, Euphorbia helioscopia, Sonchus oleraceus, Persicaria lapthifolia both in spring and summer. Chenopodium album seedlings appear in spring (in Central Europe more and more often from mid-April) and summer. The most controversial is the designation of some species as "winter". Indeed, the English language makes no distinction between “winter” weeds (species that need vernalization to flower) and “wintering” weeds (those tolerant to frost, can emerge before winter but create flowers without vernalization). Both categories are collectively referred to as “winter”. According to available data from the literature, the actual period of emergence and development of some weeds defined by the authors of the manuscript as "winter" is as follows:

Galium aparine, Descurainia sophia,  Fumaria officinalis, Lamium amplexicaule, Papaver rhoeas, Thlaspi arvense, Tripleurospermum inodorum,  Veronica polita  – both autumn and spring;

Veronica persica – spring or autumn, under good conditions all year around;

Geranium pusillum, Lamium purpureum, Capsella bursa-pastoris, Poa annua, Senecio vulgaris, Viola arvensis – nearly all year.   Perhaps additional references on weed biology would be advisable. The distribution of weeds into groups according to biology should be considered. Perhaps it is possible, for example, to create an additional  group with an emergence period covering almost the all year. A different division (classification) may change the interpretation of the results. Then it will be possible to compare the division into 4 groups of weeds based on the result of the CCA analysis (l. 192-214) and the division according to biology. Due to the new representation of biological groups, changes should be made to various sections of the manuscript, starting from the Abstract, through Results, Discussion and Conclusions. The second paragraph from the Conclusions section is more suitable for Discussion. The new weed classification according to the biology probably  will allow for a more accurate presentation of the issue contained in last paragraph. It can be expected that the dominant role in the weed infestation of field crops will be played by weeds capable of germinating regardless of the year season and weeds germinating in the warm period of spring and summer, such as Chenopodium album. However, it is not easy, because weed communities are determined by many factors, in addition to the crop rotation system and the changing climate, as well as the number and persistence of weed seeds.

In the text and in Figures 4, 5, 6, the authors use abbreviations of the Latin names of weeds, e.g. “AmaRetr”. It's not a mistake to create your own abbreviations. Since this is a scientific article in the field of herbology, it is advisable that the authors use official abbreviations for plant names, the so-called EPPO codes (formerly known as Bayer codes). Codes for cultivated and wild plants: 5 capital letters = 3 (genus) + 2 (species), e.g. „AMARE”.  

The reviewed manuscript meets the requirements for original scientific papers and after recommended corrections, may be accepted for further editorial work.

Comments on the Quality of English Language The English language is correct and understandable.
Minor stylistic and editorial corrections are recommended.

Author Response

The reviewed manuscript concerns the impact of crop rotation, tillage technology and the inclusion of catch crops on the number and species composition of weeds infesting sugar beet cultivation. The reviewed article is a valuable addition to knowledge on this issue. As expected, the conclusions are not clear, which results from the number of factors influencing weed infestation and their mutual interaction.

A: Thank you for appreciating our work and understanding the nature of the data obtained. We recognize the complexity of the factors influencing weed infestations and their interactions, which naturally leads to nuanced conclusions. Your recognition of this complexity is valuable, and we will continue to seek further research to deepen our understanding of this important issue.

The research methodology does not raise any serious concerns, but it should be detailed in section “2.3. Evaluation of weed infestation”. Information "The weed assessment was performed each spring before the application of herbicides”(l. 146-147) should be more detailed and cover the month and decade of assessment. This is a certain time interval, because the date of beet sowing, and therefore also of weed control, probably differed between years.

A: The growth stage of the sugar beet and the date of the herbicide application were essential in selecting the date for the evaluation. The specific date varies from year to year and corresponds to the weather patterns in a given year.

We have included the evaluation dates in Table 2.

The mere statement that the presence or even dominance of summer weeds was detected during the spring assessment raises doubts.

A: The classification of weed species into groups was done according to the traditional division. In the text we have added the source of the literature and the characteristics of the groups.

The advantage of the manuscript is the careful and accurate processing of the results using CCA.

A: Thank you for highlighting the strength of our manuscript, namely the careful and accurate processing of the results using Canonical Correspondence Analysis (CCA). We have taken great pains to ensure the robustness and reliability of our data analysis, and we're pleased to see this acknowledged. The use of CCA provides valuable insights into the relationships between variables, thereby enhancing the quality and depth of our findings.

The most controversial issue is the division of weeds into four groups (spring, summer, winter, perennials and others) according to biology, with particular emphasis on the germination period. Assigning weeds to the groups mentioned is not always correct.

The group of spring weeds was most correctly determined, although Polygonum aviculare seedlings develop both in spring and summer. Among the weeds identified as summer, the seeds of Chenopodium hybridum germinated rather in spring, Euphorbia helioscopia, Sonchus oleraceus, Persicaria lapthifolia both in spring and summer. Chenopodium album seedlings appear in spring (in Central Europe more and more often from mid-April) and summer.

A: The division of weed species into groups was based on the general division used in the Czech Republic. We have expanded M+M with the characteristics of the weed groups we use. However, each categorization brings some generalization.

Yes, we agree that, for example, Chenopodium album germinates from spring to summer, but it does not germinate in early spring and short-term temperatures around 0 °C damage its plants. It is therefore classified as a summer weed. The weed Polygonum aviculare germinates from early spring until summer, but temperatures around 0°C do not harm its plants. It is therefore classified as a spring weed.

The most controversial is the designation of some species as "winter". Indeed, the English language makes no distinction between “winter” weeds (species that need vernalization to flower) and “wintering” weeds (those tolerant to frost, can emerge before winter but create flowers without vernalization). Both categories are collectively referred to as “winter”. According to available data from the literature, the actual period of emergence and development of some weeds defined by the authors of the manuscript as "winter" is as follows:

Galium aparine, Descurainia sophia,  Fumaria officinalis, Lamium amplexicaule, Papaver rhoeas, Thlaspi arvense, Tripleurospermum inodorum,  Veronica polita  – both autumn and spring;

Veronica persica – spring or autumn, under good conditions all year around;

Geranium pusillum, Lamium purpureum, Capsella bursa-pastoris, Poa annua, Senecio vulgaris, Viola arvensis – nearly all year.   Perhaps additional references on weed biology would be advisable. The distribution of weeds into groups according to biology should be considered.

A: We agree with the reviewer. The group of "overwintering" weeds includes weed species that require vernalization as well as species that only overwinter. We know from available literature and practical observations that most overwintering weed species are capable of germination in both fall and spring.

Distinguishing between vernalizing and overwintering species is not always straightforward, so we have followed the traditional division and maintained a common group of overwintering weeds.

Perhaps it is possible, for example, to create an additional group with an emergence period covering almost the all year. A different division (classification) may change the interpretation of the results. Then it will be possible to compare the division into 4 groups of weeds based on the result of the CCA analysis (l. 192-214) and the division according to biology.

 A: This is a very interesting idea that we would like to use in the future. Given the heterogeneity of results and observations from only one crop.

Due to the new representation of biological groups, changes should be made to various sections of the manuscript, starting from the Abstract, through Results, Discussion and Conclusions.

A: We appreciate the reviewer's comment regarding the representation of biological groups. However, we believe it is essential for the integrity and consistency of our research to remain faithful to the traditional division of weeds into groups. While we acknowledge that changes may be necessary in various sections of the manuscript, including the Abstract, Results, Discussion, and Conclusions, we believe that maintaining consistency with established conventions in weed classification is critical for clarity and comparability across studies.

The second paragraph from the Conclusions section is more suitable for Discussion. 

A: Corrected

The new weed classification according to the biology probably will allow for a more accurate presentation of the issue contained in last paragraph.

It can be expected that the dominant role in the weed infestation of field crops will be played by weeds capable of germinating regardless of the year season and weeds germinating in the warm period of spring and summer, such as Chenopodium album. However, it is not easy, because weed communities are determined by many factors, in addition to the crop rotation system and the changing climate, as well as the number and persistence of weed seeds.

A: Weed communities are very heterogeneous and have heterogeneous responses to factors acting on them. The group of summer weeds is limited by late frosts. The occurrence of low temperatures even in late spring cannot be ruled out, and in conditions of climate change we can expect more extremes during the weather. The Chenopodium album species will still be limited by this fact, but it is possible that its importance will increase.

In the text and in Figures 4, 5, 6, the authors use abbreviations of the Latin names of weeds, e.g. “AmaRetr”. It's not a mistake to create your own abbreviations. Since this is a scientific article in the field of herbology, it is advisable that the authors use official abbreviations for plant names, the so-called EPPO codes (formerly known as Bayer codes). Codes for cultivated and wild plants: 5 capital letters = 3 (genus) + 2 (species), e.g. „AMARE”.  

A: or weed names, we used ''Kaplan et al. (2019).'' To avoid confusion between weed species, we created abbreviations based on this nomenclature.

The reviewed manuscript meets the requirements for original scientific papers and after recommended corrections, may be accepted for further editorial work.

We are pleased to inform you that we have made the recommended corrections to the manuscript. Thank you for your guidance and feedback throughout this process. We believe that these revisions have strengthened the quality and clarity of the manuscript, making it even more suitable for further editorial consideration.

Reviewer 2 Report

Comments and Suggestions for Authors

The manuscript, entitled: "The influence of sugar beet cultivation technologies on the intensity and species biodiversity of weeds", aims to answer an important question, with a complex approach.

The development of sugar beet weed control technology and the investigation of the effects of additional methods (e.g. crop rotation, cover crop technologies, tillage variants) on efficiency and weed composition are aimed at with a complex analysis. Among the factors, they also try to quantify the effect of vintages.

The objective of the analysis can result in answering serious professional questions and improving weed control efficiency. At the same time, it can be said that the authors' efforts to present this correctly were not entirely successful.

First of all, the presentation of the soil properties of the experimental area in the material and method chapter (2.1.) is not complete (e.g. Humus % and standard soil test parameters are missing). The same can be said for weather parameters. The manuscript refers to the effect of specific years on weed associations, but at the same time, the main weather parameters (at least monthly average temperature, precipitation data) broken down by year are missing.

2.2. chapter also has several gaps: the dosage of the pesticides used was not specified. The use of Convisio herbicide technology, which was not yet available in 2013, is also questionable. For the period up to 2013-2020, it therefore does not provide information regarding the herbicide technologies, nor the sugar beet varieties or hybrids used each year.

In the Results section, Figure 1 contains very small letters and is very difficult to read. Too much information is crammed into the figure, I recommend breaking it down for better clarity and readability.

In the case of Figure 3, the letter markings indicating significant differences (a-e) are missing.

When naming the weeds (e.g. lines 204-237), it is recommended to write the scientific names in italics. As it happened in other parts of the manuscript.

In view of the above, I recommend a major revision of the manuscript and resubmission.

Comments on the Quality of English Language

Minor typos need to be corrected.

Author Response

The manuscript, entitled: "The influence of sugar beet cultivation technologies on the intensity and species biodiversity of weeds", aims to answer an important question, with a complex approach.

The development of sugar beet weed control technology and the investigation of the effects of additional methods (e.g. crop rotation, cover crop technologies, tillage variants) on efficiency and weed composition are aimed at with a complex analysis. Among the factors, they also try to quantify the effect of vintages.

The objective of the analysis can result in answering serious professional questions and improving weed control efficiency. At the same time, it can be said that the authors' efforts to present this correctly were not entirely successful.

A: We appreciate your recognition of the importance of our research question and the complexity of our approach. Your feedback regarding the need for improved presentation is noted, and we have carefully revised the manuscript to improve the clarity and accuracy with which we communicate our findings. Your input is invaluable in strengthening the quality of our work, and we have done our best to address your comments to ensure that the manuscript meets the highest standards of scientific rigor.

First of all, the presentation of the soil properties of the experimental area in the material and method chapter (2.1.) is not complete (e.g. Humus % and standard soil test parameters are missing).

A: Thank you for your comment, corrected in the text.

The same can be said for weather parameters. The manuscript refers to the effect of specific years on weed associations, but at the same time, the main weather parameters (at least monthly average temperature, precipitation data) broken down by year are missing.

A: Thank you for your comment. Unfortunately, I do not have data on temperature and precipitation in the years of interest. Long-term monthly temperature and precipitation data have been added in the text.

2.2. chapter also has several gaps: the dosage of the pesticides used was not specified. The use of Convisio herbicide technology, which was not yet available in 2013, is also questionable. For the period up to 2013-2020, it therefore does not provide information regarding the herbicide technologies, nor the sugar beet varieties or hybrids used each year.

A: We appreciate your attention to detail and insightful comments. We acknowledge the gaps in the information provided, particularly with respect to pesticide dosages and the inclusion of the Convisio herbicide technology, which was not available during the specified timeframe. We have addressed these shortcomings by providing the necessary details on pesticide dosages and ensuring that only relevant technologies and practices that were available within the timeframe indicated are included in the analysis. Your feedback is invaluable in improving the quality and accuracy of our manuscript, and we are committed to making the necessary revisions to address your concerns.

In the Results section, Figure 1 contains very small letters and is very difficult to read. Too much information is crammed into the figure, I recommend breaking it down for better clarity and readability.

A: Thank you for your attention, corrected in the text.

In the case of Figure 3, the letter markings indicating significant differences (a-e) are missing.

A: Thank you for your attention, corrected in the text.

When naming the weeds (e.g. lines 204-237), it is recommended to write the scientific names in italics. As it happened in other parts of the manuscript.

A: Thank you for your attention, corrected in the text.

In view of the above, I recommend a major revision of the manuscript and resubmission.

A: Thank you for your comprehensive review and recommendation. We appreciate your thorough evaluation of the manuscript. We have carefully addressed each of the issues raised, including clarifying pesticide dosage, ensuring the accuracy of technology inclusion, and providing comprehensive information on sugar beet varieties and herbicide technologies used during the specified time period. We have undertaken a thorough review to improve quality.

Round 2

Reviewer 1 Report

Comments and Suggestions for Authors

The reviewed manuscript has been revised in accordance with the reviewers' comments. First of all, the research methodology was supplemented by adding a number of necessary information regarding the soil, the crop, protection against harmful organisms, and most importantly, the date of weed infestation assessment. This will allow the reader to better interpret the presented results. Changes in the discussion and conclusions can be considered sufficient. The revised text better indicates the role of the changing climate and agricultural practices on the species composition and number of weeds infesting sugar beet plantations. After making the changes, I recommend that the Editor accept the manuscript for publication.

Reviewer 2 Report

Comments and Suggestions for Authors

Thanks for the corrections! I recommend publishing the manuscript.
I wish the Authors more fruitful work!